# Novelties in Microthyriaceae (Microthyriales): Two New Asexual Genera with Three New Species from Freshwater Habitats in Guizhou Province, China

**DOI:** 10.3390/jof9020178

**Published:** 2023-01-28

**Authors:** Lingling Liu, Jing Yang, Si Zhou, Xiaofeng Gu, Jiulan Gou, Quanquan Wei, Meng Zhang, Zuoyi Liu

**Affiliations:** 1Guizhou Institute of Soil and Fertilizer, Guizhou Academy of Agricultural Sciences, Guiyang 550006, China; 2Guizhou Key Laboratory of Agricultural Biotechnology, Guizhou Academy of Agricultural Sciences, Guiyang 550006, China; 3Center of Excellence in Fungal Research, School of Science, Mae Fah Luang University, Chiang Rai 57100, Thailand; 4Guizhou Provincial Environmental Science Research and Design Institute, Guiyang 550081, China

**Keywords:** 5 new taxa, aquatic fungi, asexual morphs, generic delimitation, molecular phylogeny

## Abstract

Microthyriaceae is typified by the sexual genus *Microthyrium*, with eight asexual genera. Three interesting isolates were collected during our investigation of freshwater fungi from the wetlands in Guizhou Province, southwest China. Three new asexual morphs are identified. Phylogenetic analyses using ITS and LSU gene regions revealed the placement of these isolates in Microthyriaceae (Microthyriales, Dothideomycetes). Based on the morphology and phylogenetic evidence, two new asexual genera, *Paramirandina* and *Pseudocorniculariella*, and three new species, *Pa. aquatica*, *Pa. cymbiformis*, and *Ps. guizhouensis,* are introduced. Descriptions and illustrations of the new taxa are provided, with a phylogenetic tree of Microthyriales and related taxa.

## 1. Introduction

Microthyriaceae was introduced by Saccardo [1] with the sexual genus *Microthyrium* as the type genus. Microthyriaceae, poorly studied with few DNA sequence data, is the only family in Microthyriales [2,3]. Genera in the family were reappraised several times by Wu et al. [4,5,6]. They conducted the first phylogenetic analysis of Microthyriaceae using the LSU and SSU dataset and remained seven sexual genera from 50 in the family [7]. Wijayawardene et al. [8,9] accepted nine genera in the family, including two asexual genera, *Hamatispora* [10] and *Neoanungitea* [11]. Hongsanan et al. [12] added two new asexual genera and provided a holomorphic family description. Two new *Triscelophorus* species were introduced by Qiao et al. [13], and the systematic placement of *Triscelophorus* was confirmed within Microthyriaceae. Recently, four asexual genera were introduced to the family [13,14]. So far, Microthyriaceae contains 16 genera, including eight asexual genera, *Antidactylaria*, *Hamatispora*, *Isthmomyces*, *Keqinzhangia*, *Neoanungitea*, *Pseudocoronospora*, *Pseudopenidiella*, and *Triscelophorus* [12,13,14,15]. Asexual morphs in Microthyriaceae are characterized by micronematous to macronematous, mononematous, unbranched or branched, hyaline or brown conidiophores, some reduced to conidiogenous cells, integrated, terminal, determinate or sympodial, mono- to polyblastic conidiogenous cells, and subcylindrical to ellipsoid, obclavate, pale brown, verruculose, aseptate to multi-septate conidia, solitary or in branched chains, sometimes radial on compact heads; ramoconidia when present, subcylindrical to fusoid-ellipsoid, pale brown, verruculose, aseptate [12]. Sexual morphs in the family have circular, flattened, dark brown thyriothecia with radiating cells and a central ostiole, cylindrical to obpyriform, 8-spored asci and fusiform to ellipsoidal, hyaline or brown, 1-septate ascospores, often with ciliate appendages [4,8,12,13,14,15,16].

During a survey of the taxonomy and diversity of freshwater fungi from karst plateau wetlands in Guizhou Province, China [17,18,19,20,21,22], three asexual species were collected and identified based on the morphology and phylogenetic analysis. We therefore introduce two new genera, *Paramirandina* and *Pseudocorniculariella*, and three new species, *Pa*. *aquatica*, *Pa. cymbiformis*, and *Ps*. *guizhouensis*, with an illustrated account and molecular evidence. An updated backbone tree using ITS and LSU sequences is provided for Microthyriales.

## 2. Materials and Methods

### 2.1. Collection and Examination of Specimens

Specimens of submerged decaying twigs were collected from wetlands in Guizhou Province, China. Samples were taken to the laboratory in zip-lock plastic bags and incubated in plastic boxes lined with moistened sterile filter paper at room temperature for one week. Motic Nikon SMZ-171 (Nikon, Tokyo, Japan) dissecting microscopes were used to observe the fungal colonies and fruiting bodies. Fungal structures were examined and photographed using a Nikon ECLIPSE 80i (Nikon, Tokyo, Japan) compound microscope fitted with a Canon 70D (Canon, Tokyo, Japan) digital camera. Single spore isolations were made onto water agar (WA), and germinated spores were transferred onto potato dextrose agar (PDA) following the method in Luo et al. [23] and Senanayake et al. [24]. Tarosoft Image Frame Work program was used for measurement, and images used for figures were processed with Adobe Photoshop CS6 software. Herbarium specimens were deposited in the herbarium of Guizhou Academic of Agriculture Sciences (GZAAS), Guiyang, China, and herbarium of Cryptogams, Kunming Institute of Botany Academia Sinica (HKAS), Kunming, China. Axenic cultures were deposited in Guizhou Culture Collection (GZCC). Facesoffungi and Index Fungorum numbers were registered as outlined in Jayasiri et al. [25] and Index Fungorum (December 2022) [26].

### 2.2. DNA Extraction, PCR Amplification and Sequencing

Fungal mycelium was scraped using a sterilized scalpel and transferred to a 1.5 mL microcentrifuge tube for genomic DNA extraction. An Ezup Column Fungi Genomic DNA Purification Kit (Sangon Biotech, China) was used to extract DNA following the manufacturer’s instructions. DNA amplification was performed by polymerase chain reaction (PCR). ITS, LSU, SSU, *tef1-α* and *rpb2* gene regions were amplified using the primer pairs, ITS5/ITS4 [27], LR0R/LR5 [28,29], NS1/NS4 [27], *ef1*-983F/*ef1*-2218R [30], and *rpb2*-5F/*rpb2*-7cR [31,32], respectively. The amplification was performed in a 25 μL reaction volume containing 9.5 μL ddH_2_O, 12.5 μL 2 × Taq PCR Master Mix with blue dye (Sangon Biotech, China), 1 μL of DNA template, and 1 μL of each primer (10 μM). The amplification condition for LSU, ITS and *tef1-α* genes consisted of initial denaturation at 94 °C for 3 min, followed by 40 cycles of 45 s at 94 °C, 50 s at 56 °C and 1 min at 72 °C, and a final extension period of 10 min at 72 °C. The amplification condition for the *rpb2* gene consisted of initial denaturation at 95 °C for 5 min, followed by 37 cycles of 15 s at 95 °C, 50 s at 56 °C and 2 min at 72 °C, final extension period of 10 min at 72 °C. Purification and sequencing of PCR products were carried out by Shanghai Sangon Biological Engineering Technology and Services Co., Shanghai, China.

### 2.3. Phylogenetic Analyses

The ex-type and additional strains of Microthyriales species and related orders (Micropeltidales, Natipusillales, Phaeotrichale, Venturiales, and Zeloasperisporiales) were selected in the phylogenetic analyses. Two gene regions, ITS and LSU, were used for the multi-gene analyses. Sequences were optimized manually to allow maximum alignment and maximum sequence similarity. The sequences were aligned using the online multiple alignment program MAFFT v.7 (http://mafft.cbrc.jp/alignment/server/, accessed on 12 January 2023) [33]. The alignments were checked visually and improved manually where necessary.

Maximum likelihood (ML), Bayesian inference (BI), and Maximum parsimony (MP) analyses were employed to assess phylogenetic relationships. ML and BI analyses were performed through the CIPRES Science Gateway V. 3.3 [34]. ML analyses were conducted with RAxML-HPC v. 8.2.12 [35] using a GTRGAMMA approximation with rapid bootstrap analysis followed by 1000 bootstrap replicates. For the BI approach, MrModeltest2 v. 2.3 [36] was used to infer the appropriate substitution model that would best fit the model of DNA evolution for the combined dataset. The GTR+G+I substitution model was selected for ITS and LSU partitions. BI analyses were performed in a likelihood framework implemented in MrBayes 3.2.6 [37]. Six simultaneous Markov chains were run until the average standard deviation of split frequencies was below 0.01, with trees saved every 1000 generations. The first 25% of saved trees, representing the burn-in phase of the analysis, were discarded. The remaining trees were used for calculating the posterior probabilities of recovered branches [38]. MP analyses were conducted with PAUP v. 4.0a167 [39]. A heuristic search was performed with the stepwise-addition option with 1000 random taxon addition replicates and tree bisection and reconnection branch swapping. All characters were unordered and of equal weight, and gaps were treated as missing data. Maxtrees were unlimited, branches of zero length were collapsed, and all multiple, equally parsimonious trees were saved. Clade stability was assessed using a bootstrap analysis with 1000 replicates, each with ten replicates of random stepwise addition of taxa [40].

The resulting trees were printed with FigTree v. 1.4.4, and the layout was created in Adobe Illustrator 2019. Sequences generated in this study were deposited in GenBank (Table 1). 

## 3. Phylogenetic Results

Phylogenetic relationships of three Microthyriales species were assessed in the combined analysis using ITS and LSU gene regions of 54 strains in Microthyriales and related orders Micropeltidales, Natipusillales, Phaeotrichales, Venturiales, and Zeloasperisporiales. The analyzed alignment consisted of combined ITS (1–516 bp) and LSU (517–1321 bp) sequence data, including gaps. *Kirschsteiniothelia lignicola* (MFLUCC10 0036) served as outgroup taxon. The best scoring RAxML tree is shown in Figure 1. The analyzed ML, MP, and bayesian trees were similar in topology and did not conflict significantly. *Paramirandina* and *Pseudocorniculariella* formed two distinct clades and nested within Microthyriaceae. *Paramirandina aquatica* (GZCC 19-0408) grouped with *Pa. cymbiformis* (HKAS 112619) with good support (90% MLBS/0.98 PP/92% MPBS), and they formed a sister clade to *Keqinzhangia aquatica* (YMF1-04262). Comparison of the LSU sequences of *Pa. aquatica* and *Pa. cymbiformis vs. K. aquatica* showed 93.29% (57 bp different in 850 bp) and 92.88% (57 bp different in 800 bp) sequence identity, respectively. Based on the molecular data, *Pa. aquatica* differs from *Pa. cymbiformis* by 11 bp in LSU (806 bp), 39 bp in *tef1-α* (1013 bp), and 48 bp in *rpb2* (997 bp). *Pseudocorniculariella guizhouensis* (GZCC 19-0513) was resolved as a monophyletic clade with good statistical support (100% MLBS/1.0 PP/95% MPBS).

## 4. Taxonomy

### 4.1. Paramirandina *L.L. Liu & Z.Y. Liu, **gen. nov.***

Index Fungorum number: IF900034; Facesoffungi number: FoF13245

Etymology: named after its morphology similar to *Mirandina*.

*Saprobic* on decaying submerged wood in freshwater habitats. **Asexual morph:**
*Colonies* on natural substrates effuse, hairy, scattered, yellowish brown to brown, with glistening conidial masses at the apex. *Mycelium* partly superficial, partly immersed, composed of septate, brown to hyaline, smooth-walled hyphae. *Conidiophores* macronematous, mononematous, single or in small groups, unbranched, erect, straight or slightly flexuous, cylindrical, smooth-walled, multi-septate, dark brown, becoming pale brown to subhyaline towards the apex, slightly tapering towards the apex. *Conidiogenous cells* polyblastic, integrated, terminal, cylindrical to lageniform, pale brown to subhyaline, often flexuous at the apex, sometimes elongating percurrently. *Conidia* holoblastic, solitary or gathered in chains, acropleurogenous, fusiform, cymbiform or narrowly lunate, hyaline, 2–6-septate, smooth-walled, truncate at the base. **Sexual morph:** undetermined.

Type species: —*Paramirandina aquatica* L.L. Liu & Z.Y. Liu

Notes: *Paramirandina* is similar to *Heliocephala* and *Mirandina* [41,42]. *Heliocephala*, typified by *H. proliferans*, is similar to *Paramirandina* by cylindrical, brown, erect conidiophores [41]. *Paramirandina* can be distinguished from *Heliocephala* in having integrated, polyblastic conidiogenous cells and cymbiform, fusiform or narrowly lunate conidia. *Heliocephala* has monoblastic, discrete conidiogenous cells and obclavate and rostrate conidia. *Mirandina*, typified by *M. corticola*, is characterized by brown, erect conidiophores, cylindrical, polyblastic conidiogenous cells, and hyaline, clavate, filiform or fusiform conidia, usually with short-cylindrical denticles in apical clusters [42]. However, the short-cylindrical denticles are absent in *Paramirandina*. Phylogeneticanalysis showed that *Paramirandina* belongs to Microthyriales, while *Mirandina* belongs to Helotiales [26]. *Paramirandina* shares the morphology with *Pleurotheciella* and *Pleurothecium* in having macronematous brown conidiophores, polyblastic conidiogenous cells and hyaline conidia. However, conidiogenous cells of *Pleurotheciella* and *Pleurothecium* are with minute or tooth-like denticles. *Pleurotheciella* and *Pleurothecium* are members of Pleurotheciaceae (Pleurotheciales, Sordariomycetes) [43]. In our phylogenetic analysis, *Paramirandina* is sister to *Keqinzhangia aquatica* (Figure 1). *Paramirandina* has cymbiform, fusiform or narrowly lunate conidia, while *Keqinzhangia* has cylindrical, obclavate, bacilliform, fusiform, sub-oblecythiform or cuneiform conidia [14]. Comparison of the LSU sequences of *Pa. aquatica* and *K. aquatica* showed 93.29% (793/850 bp) identity, while *Pa. cymbiformis* and *K. aquatica* showed 92.88% (743/800 bp) identity.

*Paramirandina aquatica* L.L. Liu & Z.Y. Liu, sp. nov., Figure 2.

Index Fungorum number: IF900038; Facesoffungi number: FoF13246

Etymology: referring to the aquatic habitat of the species

Holotype: GZAAS 20-0303

*Saprobic* on decaying submerged wood in freshwater habitats. **Asexual morph:**
*Colonies* on natural substrates effuse, hairy, scattered, yellowish brown, with glistening conidial masses at the apex. *Mycelium* partly superficial, partly immersed, composed of septate, brown to hyaline, smooth-walled hyphae. *Conidiophores* macronematous, mononematous, solitary or in small groups, unbranched, erect, straight or slightly flexuous, cylindrical, smooth-walled, 6–10-septate, brown, becoming pale brown to subhyaline and tapering towards the apex, 138–200 × 4.5–8 μm ($\bar{X\;}$= 171 × 6 μm, *n* = 15). *Conidiogenous cells* polyblastic, integrated, terminal, determinate, sympodial, cylindrical to lageniform, pale brown to subhyaline, often flexuous at the apex, sometimes elongating percurrently. *Conidia* holoblastic, solitary or gathered in chains, acropleurogenous, cymbiform, fusiform or narrowly lunate, obtuse at the apex, truncate at the base, sometimes slightly curved, hyaline, 2–5-septate, mostly 4-septate, 23–34 × 4–7.5 μm ($\bar{X\;}$= 28 × 6 μm, *n* = 30), smooth, sometimes bearing a new conidium at the apex. **Sexual morph:** undetermined.

Culture characteristics: Conidia germinating on WA medium within 24 h and germ tube produced from one or both ends. Colonies on PDA medium slow growing, reaching about 10 mm diam. after two months at 25 °C in dark, circular, with dense, velvety, grayish white to brown mycelium on the surface; in reverse dark brown to black with entire margin.

Material examined: CHINA, Guizhou Province, Dushan District, deep ditch scenic spot, near 25°55′N, 107°37′E, at an altitude of 1205m, on decaying branch submerged in a stream, 5 July 2018, L.L. Liu, 18D-66 (GZAAS 20-0303, **holotype**), ex-type culture GZCC 19-0408; additional sequences, SSU: OQ025204; *tef1-α*: OQ032664; *rpb2*: OQ032662

Notes: *Paramirandina aquatica* is similar to *Heliocephala variabilis* [41] in conidiophores; similar to *Mirandina inaequalis* [44] and *Keqinzhangia aquatica* [14] in conidial shape; and similar to *Pa. cymbiformis* (HKAS 112619). However, *Pa. aquatica* differs from *H. variabilis* by cymbiform or narrowly lunate, fusiform conidia, and polyblastic conidiogenous cells. *Paramirandina aquatica* is distinguished from *M. inaequalis* in lacking the short-cylindrical denticles in the upper region of the conidiogenous cells. *Keqinzhangia aquatica* was described on culture. The conidial shape and size of *Pa. aquatica* on natural substrate differs from *K*. *aquatica* on culture. Conidia of *Pa. aquatica* are shorter than that of *K*. *aquatica* (23–34 μm vs. 12–76.5 μm). *Paramirandina aquatica* has mostly cymbiform, 4-septate conidia while *K*. *aquatica* has narrowly fusiform, 0–6(–7)-septate, conidia with acute ends. *Paramirandina aquatica* (GZCC 19-0408) and *K*. *aquatica* (YMF1-04262) showed 93.29% (793/850 bp) sequence identity of the LSU gene region. *Paramirandina aquatica* shares the similar morphology with *Pa. cymbiformis* but differs by shorter conidiophores (138–200 µm vs. 280–350 µm). Comparisons of the LSU, *tef1-α,* and *rpb2* sequences of *Pa. aquatica* (GZCC 19-0408) and *Pa. cymbiformis* (HKAS 112619) showed 11 bp differences in LSU, 39 bp in *tef1-α*, and 48 bp in *rpb2* gene regions, respectively.

*Paramirandina cymbiformis* J. Yang & Z.Y. Liu, **sp. nov.**, Figure 3.

Index Fungorum number: IF900037; Facesoffungi number: FoF13247

Etymology: referring to the cymbiform conidia.

Holotype: HKAS 112619

*Saprobic* on decaying submerged wood in freshwater habitats. **Asexual morph:**
*Colonies* on wood effuse, hairy, scattered, brown, with glistening conidial masses at the apex. *Mycelium* partly superficial, partly immersed, composed of septate, smooth-walled, brown to hyaline hyphae. *Conidiophores* macronematous, mononematous, erect, straight or slightly flexuous, solitary, cylindrical, smooth-walled, septate, unbranched, dark brown, becoming pale brown to subhyaline towards the apex, 280–350 × 5–9 µm ($\bar{X}$ = 313 × 7 µm, *n* = 20). *Conidiogenous cells* polyblastic, integrated, terminal, determinate, sympodial, cylindrical, pale brown to subhyaline, often flexuous at the apex, elongating percurrently. *Conidia* acropleurogenous, aggregated in slimy masses, solitary, cymbiform or narrowly lunate, 3–6-septate, smooth-walled, hyaline, 24–30 × 5–6.5 µm ($\bar{X}$ = 26.5 × 5.5 µm, *n* = 30), guttulate, thin-walled, sometimes slightly constricted at the septa. **Sexual morph**: Undetermined.

Material examined: CHINA, Guizhou Province, Chishui City, Sidonggou Waterfall, 25°27.38′ N, 107°39.85′ E, on decaying twig submerged in a freshwater stream, 11 July 2019, J. Yang, CS 53-1 (HKAS 112619, **holotype**; HKAS 125927, isotype); additional sequence, SSU: OQ025205; *tef1-α*: OQ032665; *rpb2*: OQ032663

Notes: *Paramirandina cymbiformis* share the similar morphology with *Pa. aquatica* except for the dimension of conidiophores. However, they are distinct species based on the molecular data. 

Attempts to preserve the living culture were unsuccessful since few conidia germinated, with no growth after reaching 1–2 mm diam..

### 4.2. Pseudocorniculariella *L.L. Liu & Z.Y. Liu, **gen. nov.***

Index Fungorum number: IF900035; Facesoffungi number: FoF13248

Etymology: referring to the morphology similar genus *Corniculariella*.

*Saprobic* on decaying submerged twigs in freshwater habitats. **Asexual morph:**
*Conidiomata* effuse, sporodochial, synnematous or absent, solitary to gregarious, dark brown to black, stromatic, obpyriform, subcylindrical to subconical, slightly swollen at the base or level of the locule, narrower towards the apex, scattered over the substrate surface, minutely scabrous, reticular. *Conidiomatal wall* composed of closely interwoven septate hyphae, compacted towards exterior, dark brown to black cells of *textura angularis*, becoming thin-walled and hyaline toward the inner region. *Conidiophores* hyaline, cylindrical, branched, developed from the inner layer of the conidiomata, reduced to conidiogenous cells. *Conidiogenous* cells hyaline, enteroblastic, polyphialidic, subcylindrical or cylindrical to ampulliform, indeterminate, forming conidia at their tips, discrete or integrated, smooth, moderately thick-walled. *Conidia* solitary, hyaline, smooth, guttulate to granular, septate, slightly constricted at septa, thick-walled, filiform, acerose, tapering towards both ends, slightly curved, base truncate. **Sexual morph:** Undetermined.

Type species: —*Pseudocorniculariella guizhouensis* L.L. Liu & Z.Y. Liu, **sp. nov.**

Notes: Phylogenetic study based on ITS and LSU sequence data showed that *Pseudocorniculariella* formed a separate branch in Microthyriaceae (Microthyriales) close to *Isthmomyces* (Figure 1). *Pseudocorniculariella* is distinct from *Isthmomyces* in the formation of conidiogenous cells and conidia. *Pseudocorniculariella* has enteroblastic and polyphialidic conidiogenous cells and hyaline, filiform or falcate conidia. *Isthmomyces* has polyblastic conidiogenous cells, and two cellular isthmic-segment obclavate, clavate, pyriform conidia [13]. *Pseudocorniculariella* shares similar characteristics with *Corniculariella* (Dermateaceae, Medeolariales, Leotiomycetes) in having subconical conidiomata and hyaline, filiform conidia. However, it can be distinguished by stromatic conidiomata and polyphialidic conidiogenous cells. Phylogenetic analysis showed that *Pseudocorniculariella* belongs to Microthyriales, Dothideomycetes, while *Corniculariella* belongs to Medeolariales, Leotiomycetes [45]. Based on the morphology and phylogeny, *Pseudocorniculariella* is introduced as a new genus in Microthyriaceae. Additional collections and further molecular evidence are needed to confirm its taxonomy.

*Pseudocorniculariella guizhouensis* L.L. Liu & Z.Y. Liu, **sp. nov.**, Figure 4.

Index Fungorum number: IF900036; Facesoffungi number: FoF13249

Etymology: referring to the collecting site in Guizhou Province, China.

Holotype: GZAAS 20-0408

*Saprobic* on decaying submerged twigs in freshwater habitats. **Asexual morph:**
*Conidiomata* effuse, sporodochial, synnematous or absent, solitary to gregarious, dark brown to black, stromatic, obpyriform, subcylindrical to subconical, slightly swollen at the base or level of the locule, narrower towards the apex, scattered over the substrate surface, minutely scabrous, reticular, 65–90 µm diam., 135–175 µm high, uninoculated, reticular thin-walled, papillate. *Conidiomatal wall* composed of closely interwoven septate hyphae, compacted towards the exterior, dark brown cells of *textura angularis*, becoming thin-walled, up to 2 µm wide, and hyaline toward the inner region. *Conidiophores* formed from the inner wall of the conidiomata, reduced to conidiogenous cells. *Conidiogenous cells* polyphialidic, hyaline, definite, smooth, subcylindrical to ampulliform, 9.5–12 × 3.5–4.5 µm ($\bar{X}$ = 10.8 × 4.2 μm, *n* = 20). *Conidia* solitary, hyaline, 6–8-septate, mostly 7-septate, slightly constricted at septa, smooth, guttulate to granular, filiform, acerose, base truncates, 55.5–76.5 × 2.5–4.0 μm ($\bar{X}$ = 68.5 × 3.2 μm, *n* = 20). **Sexual morph:** Undetermined.

Culture characteristics: Conidia germinating on PDA medium within 24 h and germ tubes produced from both ends. Colonies growing on PDA medium slow growing, reaching 8–10 mm in three weeks at 25 °C in natural light, circular, with dense, olive-green mycelium in the middle, darker of the inner ring, with sparser, brown mycelium of the outer ring on the surface, in reverse dark brown to black with irregular margin.

Material examined: CHINA, Guizhou Province, Aha Lake, 26°32′ N, 106°40′ E, at an altitude of 1085 m, on decaying submerged twigs in the lake, 16 April 2018, L.L. Liu, 18A-14 (GZAAS 20-0408, **holotype**), ex-type culture GZCC 19-0513; additional sequence, *tef1-α*: OQ032666

Notes: *Pseudocorniculariella guizhouensis* resembles *Corniculariella rhamni* in possessing subconical conidiomata and hyaline, filiform, conidia [45]. However, *Ps. guizhouensis* possesses stromatic conidiomata and polyphialidic conidiogenous cells, while *C. rhamni* has monophialidic conidiomata and phialidic conidiogenous cells. Phylogenetic analyses based on ITS and LSU showed that *Ps. guizhouensis* (GZCC 19-0513) belongs to Microthyriales, Dothideomycetes, while *Corniculariella* is a member of Medeolariales, Leotiomycetes.

## 5. Discussion

Freshwater fungi are a heterogeneous group. With the increasing abundance of molecular data, the numbers have rapidly increased [46,47,48,49,50,51,52,53]. According to statistics, there are 3,870 freshwater fungal species [54]. They mainly consist of Ascomycota (Sordariomycetes, Dothideomycetes, Eurotiomycetes, Leotiomycetes) and other phyla, including Chytridiomycota, Basidiomycota and Rozellomycota species. The Ascomycota accounted for three-quarters of the total [54].

In recent years, the molecular phylogeny of freshwater fungi has been updated several times [48,49,54]. Nevertheless, Microthyriaceae (Microthyriales) was omitted, although a freshwater genus *Hamatispora* has been reported before then [10]. Recently, four new freshwater genera, *Antidactylaria*, *Isthmomyces*, *Keqinzhangia*, and *Pseudocoronospora*, were reported [10,11]. In this study, the combined ITS and LSU tree (Figure 1) showed that three new isolates formed two clades in Microthyriaceae. Based on the morphology and molecular evidence, we establish two new asexual genera and three new species, named *Paramirandina* and *Pseudocorniculariella,* with *Pa. aquatica*, *Pa. cymbiformis* and *Ps. guizhouensis*. Eight freshwater genera are known in Microthyriaceae, including the two new genera in this study. It is worth noting that both are asexual genera. In the previous studies about freshwater fungi, few aquatic asexual genera were included [55]. The new genus *Paramirandina* is phylogenetically close to the asexual genus *Keqinzhangia* (Figure 1)*. Paramirandina* is well distinguishable from *Keqinzhangia* by relatively long conidiophores (more than 150 µm long vs. prostrate, not differentiated), conidiogenesis (holoblastic vs. holothallic) and the conidial shape (cymbiform or narrowly lunate vs. cylindrical, cylindrical-obclavate, obclavate, bacilliform) [12]. The sequence identity of the LSU gene region between two *Paramirandina* species and *K*. *aquatica* is relatively low (93.29% and 92.88%). Therefore, we introduce *Paramirandina* as a new genus.

*Pseudocorniculariella* is phylogenetically close to the asexual genus *Isthmomyces* (Figure 1). It is distinguished from *Isthmomyces* by the formation of conidiogenous cells and the morphology of conidia, and was identified as a new genus. 

Microthyriaceae has ever been a poorly studied group. However, its taxonomic studies have greatly advanced with molecular data resulting in a rapid increase of genera numbers. Formerly, only seven genera were accepted in the family by Wu et al. [7], nine were accepted by Wijayawardene et al. [8,9], 11 were accepted by Hongsanan et al. [12], and 18 were accepted at present (this study). However, most genera contained fewer species, such as eight monotypic genera, *Chaetothyriothecium*, *Hamatispora*, *Keqinzhangia, Paramicrothyrium*, *Pseudomicrothyrium*, *Pseudocoronospora, Tumidispora*, and the new genus *Pseudocorniculariella*, and four genera with two or three species. Thus, more collections and further molecular evidence are needed to confirm the taxonomy of these genera.

Furthermore, the taxonomy of earlier proposed genera needs to be confirmed by molecular data. *Heliocephala* is the first hyphomycetous genus described in Microthyriales. There are eight species in the genus, five of which have available molecular DNA data without the type species *H. proliferans* [56,57]. Gonzalez et al. accepted *Heliocephala* in Microthyriaceae based on *H. variabilis* [57]. However, Calabon et al. [54] referred *Heliocephala* to Microthyriales *incertae sedis.* Thus, the taxonomy of *Heliocephala* needs to be reappraised with molecular DNA data of the type species and more collections.

## Figures and Tables

**Figure 1 jof-09-00178-f001:**
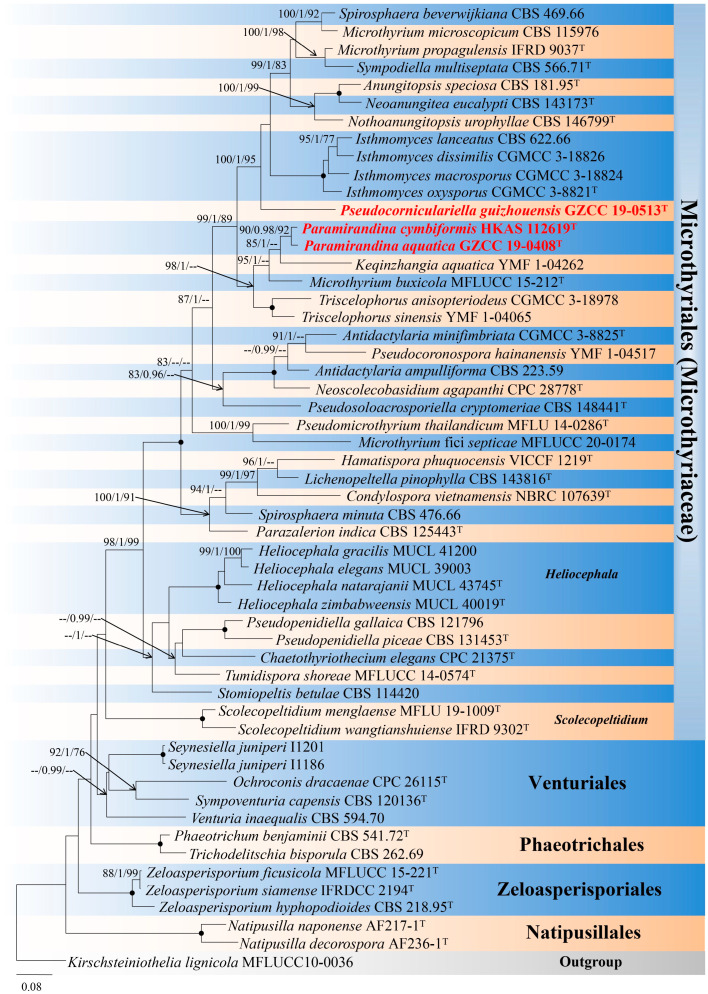
Maximum likelihood majority rule consensus tree for Microthyriales and related orders using ITS and LSU sequence data. Bootstrap support values for maximum likelihood (ML) and maximum parsimony (MP) greater than 75% and Bayesian posterior probabilities greater than 0.95 are indicated above branches as ML BS/PP/MP BS. The scale bar represents the expected number of changes per site. The tree is rooted with *Kirschsteiniothelia lignicola* (MFLUCC 10-0036). Ex-type strains are indicated with T. The new taxa are in red bold. Branches with 100% ML BS, 1.0PP and 100% MP BS were dotted with black dots. Orders are indicated as colored blocks.

**Figure 2 jof-09-00178-f002:**
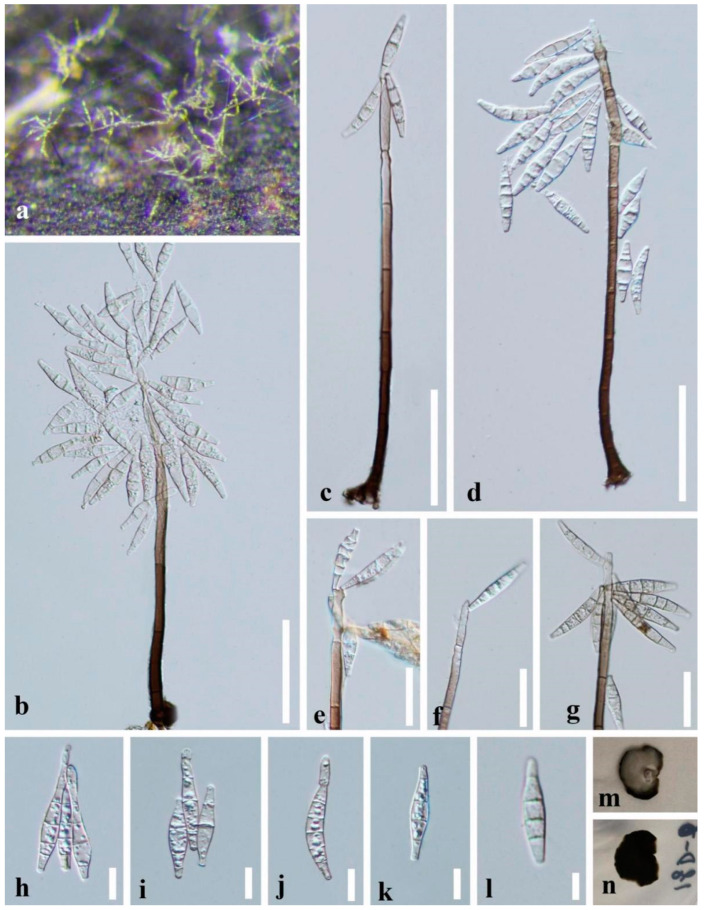
*Paramirandina aquatica* (GZAAS 20-0303, **holotype**). (**a**) Colony on wood. (**b**–**d**) Conidiophores and conidia. (**e**–**g**) Conidiogenous cells and conidia. (**h**–**l**) Conidia. (**m**,**n**) Culture, m from above, n from below. Scale bars: (**b**–**d**) = 50 μm, (**e**–**g**) = 25 μm, (**h**–**l**) = 10 μm.

**Figure 3 jof-09-00178-f003:**
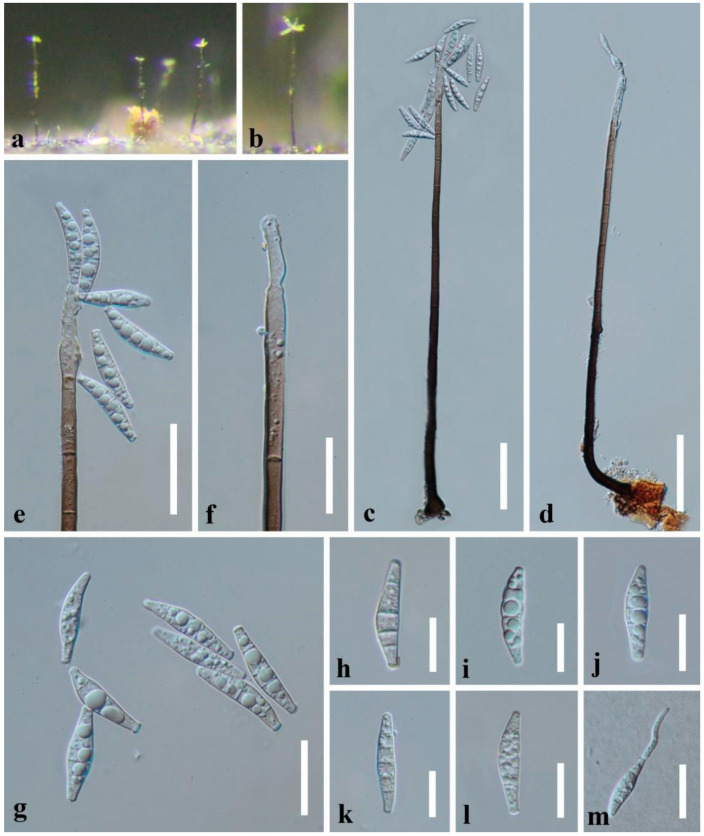
*Paramirandina cymbiformis* (HKAS 112619, **holotype**) (**a**,**b**) Colony on wood. (**c**,**d**) Conidiophores with conidia. (**e**) Conidiogenous cell with conidia. (**f**) Conidiogenous cell. (**g**–**l**) Conidia. (**m**) Germinated conidium. Scale bars: (**c**,**d**) = 50 µm, (**e**) = 30 µm, (**f**,**g**,**m**) = 20 µm, (**h**–**l**) = 15 µm.

**Figure 4 jof-09-00178-f004:**
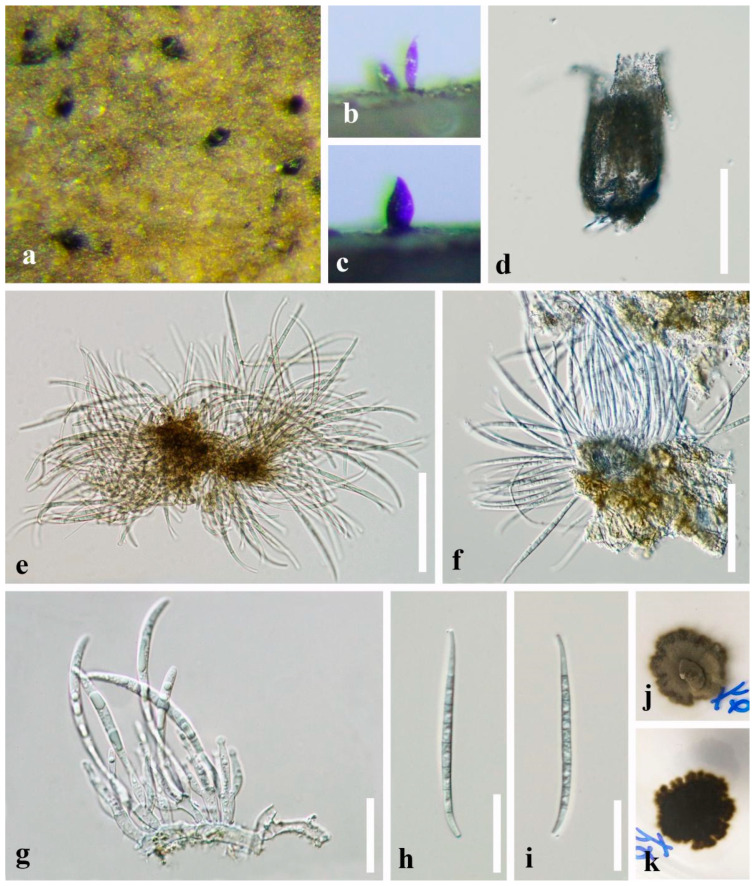
*Pseudocorniculariella guizhouensis* (GZAAS20-0408, **holotype**) (**a**–**c**) Conidiomata on wood. (**d**) Pycnidium. (**e**–**g**) Conidiogenous cells and conidia. (**h**,**i**) Conidia. (**j**,**k**) Culture, j from above, k from below. Scale bars: (**d**) =100 μm, (**e**,**f**) =50 μm, (**g**–**i**) =25 μm.

**Table 1 jof-09-00178-t001:** Taxa used in the phylogenetic analyses and their GenBank accession numbers. T denotes ex-type strains. Newly generated sequences are in bold.

Taxon	Voucher/Strain Number	GenBank Accession Number	
		LSU	ITS
*Antidactylaria ampulliforma*	CBS 223.59	MH869386	MH857845
*Antidactylaria minifimbriata*	CGMCC 3-18825^T^	MK577808	MK569506
*Anungitopsis speciosa*	CBS 181.95^T^	EU035401	EU035401
*Chaetothyriothecium elegans*	CPC 21375^T^	KF268420	-
*Condylospora vietnamensis*	NBRC 107639^T^	LC146725	LC146723
*Hamatispora phuquocensis*	VICCF 1219^T^	LC064073	LC064074
*Heliocephala elegans*	MUCL 39003	HQ333478	HQ333478
*Heliocephala gracilis*	MUCL 41200	HQ333479	HQ333479
*Heliocephala natarajanii*	MUCL 43745^T^	HQ333480	HQ333480
*Heliocephala zimbabweensis*	MUCL 40019^T^	HQ333481	HQ333481
*Isthmomyces dissimilis*	CGMCC 3 18826	MK577811	MF740794
*Isthmomyces lanceatus*	CBS 622.66	MH870563	MH858897
*Isthmomyces macrosporus*	CGMCC 3-18824	MK577812	MF740796
*Isthmomyces oxysporus*	CGMCC 3-18821^T^	MK577810	MF740793
*Keqinzhangia aquatica*	YMF 1-04262	MK577809	MK569507
*Kirschsteiniothelia lignicola*	MFLUCC 10-0036	HQ441568	HQ441567
*Lichenopeltella pinophylla*	CBS 143816T	MG844152	-
*Microthyrium buxicola*	MFLUCC 15-0212^T^	KT306551	-
*Microthyrium fici-septicae*	MFLUCC 20-0174^T^	MW063252	-
*Microthyrium microscopicum*	CBS 115976	GU301846	-
*Microthyrium propagulensis*	IFRD 9037^T^	KU948989	-
*Natipusilla decorospora*	AF236-1^T^	HM196369	-
*Natipusilla naponense*	AF217-1^T^	HM196371	-
*Neoanungitea eucalypti*	CBS 143173^T^	MG386031	MG386031
*Neoscolecobasidium agapanthi*	CPC 28778^T^	NG_059748	NR_152546
*Nothoanungitopsis urophyllae*	CBS 146799^T^	MW883825	MW883433
*Ochroconis dracaenae*	CPC 26115^T^	KX228334	KX228283
** *Paramirandina aquatica* **	**GZCC 19-0408^T^**	**OQ025201**	**OQ025199**
** *Paramirandina cymbiformis* **	**HKAS 112619^T^**	**OQ025202**	-
*Parazalerion indica*	CBS 125443^T^	MH874977	MH863483
*Phaeotrichum benjaminii*	CBS 541.72^T^	MH872266	MH860561
** *Pseudocorniculariella guizhouensis* **	**GZCC 19-0513^T^**	**OQ025203**	**OQ025200**
*Pseudocoronospora hainanensis*	YMF 1-04517	MK577807	MK569505
*Pseudomicrothyrium thailandicum*	MFLU 14-0286^T^	MT741680	-
*Pseudopenidiella gallaica*	CBS 121796	LT984843	LT984842
*Pseudopenidiella piceae*	CBS 131453^T^	JX069852	JX069868
*Pseudosoloacrosporiella cryptomeriae*	CBS 148441^T^	NG_081320	NR_175206
*Scolecopeltidium menglaense*	MFLU 19-1009^T^	MW003710	MW003724
*Scolecopeltidium wangtianshuiense*	IFRD 9302^T^	NG_067860	NR_166263
*Seynesiella juniperi*	I1201	MW405232	MW405223
*Seynesiella juniperi*	I1186	MW405230	MW405222
*Spirosphaera beverwijkiana*	CBS 469.66	HQ696657	HQ696657
*Spirosphaera minuta*	CBS 476.66	HQ696659	HQ696659
*Stomiopeltis betulae*	CBS 114420	GU214701	GU214701
*Sympodiella multiseptata*	CBS 566.71^T^	MH872028	MH860264
*Sympoventuria capensis*	CBS 120136^T^	KF156104	DQ885906
*Trichodelitschia bisporula*	CBS 262.69	MH871039	MH859305
*Triscelophorus anisopteriodeus*	CGMCC 3-18978	MK577818	MK569511
*Triscelophorus sinensis*	YMF 1-04065	MK577820	MK569513
*Tumidispora shoreae*	MFLUCC 14-0574^T^	KT314074	-
*Venturia inaequalis*	CBS 594.70	GU301879	KF156040
*Zeloasperisporium ficusicola*	MFLUCC 15-0221^T^	KT387733	-
*Zeloasperisporium hyphopodioides*	CBS 218.95^T^	EU035442	EU035442
*Zeloasperisporium siamense*	IFRDCC 2194^T^	JQ036228	-

## Data Availability

The datasets generated for this study can be found in the NCBI database.

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
