# Peer review of "Novelties in Microthyriaceae (Microthyriales): Two New Asexual Genera with Three New Species from Freshwater Habitats in Guizhou Province, China"

_jof, 2023, doi:10.3390/jof9020178_

Round 1
Reviewer 1 Report
The manuscript is well written and the information is complete, clear and straightforward in material and methods. The illustrations are very good and sufficient to understand the morphology of each species.
In the attached file there are minor formatting corrections.
However, I had some doubts regarding the differentiation of the two species of Paramirandina (observation number 2 below).
1) It was not clear to me why Pseudocorniculariella guizhouensis needs additional collections and further molecular evidence to confirm its taxonomy. (lines 316-317 and 393-394).
2) I am not so convinced that Paramirandina cymbiformis and P. aquatica are different species. Support in phylogenetic analysis is low (87/0.98/90) and morphological difference is questionable (size of conidiophore and solitary conidia in P. cymbiformis).

Author Response
The revisions responsed to you are indicated as bright blue. The revisions responsed to editor are indicated as bright yellow

Reviewer 2 Report
This is an interesting paper. Two new genera and three new species were introduced in this study. The descriptions of the new taxa were overall good. However, there are still some deficiencies. Some suggestions are as follows:
1. I suggest including the related genera (Lichenopeltella, Nothoanungitopsis, Pseudomicrothyrium, Scolecopeltidium, Seynesiella, Neoscolecobasidium, Parazalerion, Pseudosoloacrosporiella) in the phylogentic tree.
2. I suggest marking the Microthyriaceae clade on the phylogenetic tree.
3. I suggest marking the clades of different genera in Microthyriaceae on the phylogenetic tree.
4. I would like the see the morphological comparison of Paramirandina to Pleurothecium,Pleurotheciella, which are two typical genera of freshwater anamorphic genera.
5. There are many writing probems, for examples:
Line 18:from the wetlands.
Line 158: The new taxa are in bold and rad rad - red.
Line 178-179: Repetitive expression
Author Response
The revisions are indicated as red.

Reviewer 3 Report
it is a very good writing paper and well organized, if the authors can improve their English editing would be much better for readers.

Author Response
The revisions responsed to you are indicated as bright green. The revisions responsed to editor are indicated as bright yellow

Round 2
Reviewer 2 Report
No more comments